# The time to peak blood bicarbonate (HCO₃⁻), pH, and the strong ion difference (SID) following sodium bicarbonate (NaHCO₃) ingestion in highly trained adolescent swimmers

**Josh W. Newbury[1], Matthew Cole[1], Adam L. Kelly[1], Richard J. Chessor[2], S. Andy Sparks[3]\*, Lars R. McNaughton[3], Lewis A. Gough[1]**

**1** Human Performance and Health Research Group, Centre for Life and Sport Sciences (CLaSS), Department of Sport and Exercise, Birmingham City University, Birmingham, United Kingdom, **2** Sports Science and Sports Medicine Team, British Swimming, Loughborough, Leicestershire, United Kingdom, **3** Sports Nutrition and Performance Research Group, Department of Sport and Physical Activity, Edge Hill University, Ormskirk, United Kingdom

\* andy.sparks@edgehill.ac.uk

**Data Availability Statement:** We have now uploaded the data set as a S1 Data.

## Abstract

The timing of sodium bicarbonate (NaHCO₃) supplementation has been suggested to be most optimal when coincided with a personal time that bicarbonate (HCO₃⁻) or pH peaks in the blood following ingestion. However, the ergogenic mechanisms supporting this ingestion strategy are strongly contested. It is therefore plausible that NaHCO₃ may be ergogenic by causing beneficial shifts in the strong ion difference (SID), though the time course of this blood acid base balance variable is yet to be investigated. Twelve highly trained, adolescent swimmers (age: 15.9 ± 1.0 years, body mass: 65.3 ± 9.6 kg) consumed their typical pre-competition nutrition 1–3 hours before ingesting 0.3 g·kg BM⁻¹ NaHCO₃ in gelatine capsules. Capillary blood samples were then taken during seated rest on nine occasions (0, 60, 75, 90, 105, 120, 135, 150, 165 min post-ingestion) to identify the time course changes in HCO₃⁻, pH, and the SID. No significant differences were found in the time to peak of each blood measure (HCO₃⁻: 130 ± 35 min, pH: 120 ± 38 min, SID: 98 ± 37 min; p = 0.08); however, a large effect size was calculated between time to peak HCO₃⁻ and the SID (g = 0.88). Considering that a difference between time to peak blood HCO₃⁻ and the SID was identified in adolescents, future research should compare the ergogenic effects of these two individualized NaHCO₃ ingestion strategies compared to a traditional, standardized approach.

## Introduction

Sodium bicarbonate (NaHCO₃) is recommended to athletes based upon strong evidence of a performance enhancing effect [1]. It is well acknowledged that NaHCO₃ ingestion enhances buffering capacity by increasing bicarbonate (HCO₃⁻) and pH concentrations within the

**Funding:** RJC contributed to this research by providing peer review from the perspective of a world-class performance nutritionist. British Swimming as an organisation had no role within any aspect of this research project. All funding and consumables were provided by Birmingham City University, and all participants were volunteers from a local high-performance swimming team (City of Birmingham Swimming Club).

**Competing interests:** The commercial affiliation of RJC does not alter our adherence to PLOS ONE policies on sharing data and materials.

blood, although the magnitude of these increases is variable between individuals [2, 3]. Increasing blood alkalosis alters the pH gradient between the intracellular and extracellular compartments, subsequently leading to the upregulation of the lactate–hydrogen ion (H$^+$) co-transporter to efflux acidic H$^+$ from the active musculature and into circulation [4, 5]. During exercise, accelerating the removal of H$^+$ is purported to offset fatigue since intracellular acidosis is associated with debilitating effects on the capacity to sustain muscle force production [6–10]. Specifically, cellular acidosis is suggested to inhibit muscle shortening velocity and cross-bridge cycling by reducing calcium ion (Ca$^{2+}$) sensitivity and myosin ATPase activity [10, 11]. Further impairments of key glycolytic enzymes [12, 13] and the strong ion difference (SID) [7, 14, 15] are also associated with exacerbated H$^+$ accumulation, hence limiting the available ATP substrates and action potentials necessary to maintain muscular contractions, respectively. However, each of these fatiguing mechanisms are strongly contested [16–18]. Despite this, NaHCO$_3$ continues to provide an ergogenic benefit to exercise performance, subsequently requiring further investigation of the biochemical changes that occur following ingestion.

There is an apparent increase in the ergogenic potential of NaHCO$_3$ when baseline blood HCO$_3^-$ is increased by +5 mmol·L$^{-1}$, whereas increases above +6 mmol·L$^{-1}$ are associated with almost certain performance enhancement [19, 20]. However, the time taken to reach these thresholds varies considerably between individuals when NaHCO$_3$ is consumed in either capsule (range: 40–240 min [3, 21, 22]) or solution form (range: 40–125 min [2, 23–25]), therefore highlighting a potential flaw in the current dosing guidelines (i.e., 60–150 min pre-exercise for all athletes [1]). Contemporary research is addressing this issue by timing exercise to coincide with an individual peak blood HCO$_3^-$ concentration [21, 23–25], though at present, only Boegman et al. [21] has directly compared exercise performance between NaHCO$_3$ ingested at time to peak HCO$_3^-$ (40–160 min pre-exercise) versus a traditional dosing protocol (60 min pre-exercise for all athletes). The individualized approach significantly improved the rowing performance (2000 m time-trial) of 18 of 23 world-class rowers (Individualized = 367.0 ± 10.5 s vs. Traditional = 369.0 ± 10.3 s), however, this result occurred with only a small difference in pre-exercise HCO$_3^-$ (Individualized = +6 mmol·L$^{-1}$ vs. Traditional = +5.5 mmol·L$^{-1}$). It is therefore plausible that another mechanism besides those associated with increased HCO$_3^-$ is involved with the ergogenic properties of NaHCO$_3$ supplementation.

Alternatively, NaHCO$_3$ may mitigate fatigue by altering the intracellular and extracellular balance of strong ions such as potassium (K$^+$), sodium (Na$^+$), chloride (Cl$^-$), and Ca$^{2+}$ [26]. Specifically, NaHCO$_3$ increases the influx of K$^+$ into the muscle following ingestion and attenuates losses in intramuscular K$^+$ during exercise, subsequently improving the capacity to sustain muscle excitability [27–30]. A concomitant increase in muscular Cl$^-$ uptake is also observed post-ingestion [27, 29, 30], which is suspected to work synergistically with K$^+$ to protect force and excitability when muscles begin to depolarize [31]. These effects are further strengthened by an increased plasma Na$^+$ [2, 3, 27, 28], which together with changes in K$^+$ and Cl$^-$, could indicate an upregulation of Na$^+$/K$^+$–ATPase and Na$^+$/K$^+$/2Cl$^-$–ATPase activity to limit depolarization and preserve excitation-contraction coupling following NaHCO$_3$ ingestion [16, 30, 31]. Currently, only Gough et al. have reported changes in the collective SID following NaHCO$_3$ ingestion and whole-body exercise [24, 27]. Interestingly, both studies identified an ergogenic effect of NaHCO$_3$ (4 km cycling time-trial: 1.4% faster [24]; treadmill time-to-exhaustion: 28% further [27]) when blood HCO$_3^-$ increases (+7.0 mmol·L$^{-1}$ [24]; +8.8 mmol·L$^{-1}$ [27]) were identified alongside an increased SID (+6 mEq·L$^{-1}$ [24]; +10 mEq·L$^{-1}$ [27]) prior to exercise. Given that the role of H$^+$ accumulation is controversial within exercise fatigue [16–18], the resultant performance improvements could instead be attributed to an increased plasma SID. However, no study to date has purposely intended to identify a peak

SID measurement. If a time difference exists between the time taken to reach peak blood HCO$_3^-$ and the SID concentrations, then it is plausible that future NaHCO$_3$ dosing strategies based upon a time to peak SID could optimize the SID mechanisms associated with performance enhancement.

Adolescence (age: 15–19 years) is a critical time in an athlete's career, whereby competitive success is essential for continued sports participation [32]. Young athletes are therefore attracted to nutritional ergogenic aids to support their performance, though in practice, these are not always ingested in an evidence-based fashion [33]. Moreover, whether NaHCO$_3$ supplementation is appropriate for trained adolescents is unclear, with there being little evidence to suggest that large increases in blood alkalosis occur following ingestion. For example, neither Zajac et al. (+3.4 mmol·L$^{-1}$ [34]) or Guimarães et al. (+3.4 mmol·L$^{-1}$ [35]) observed a blood HCO$_3^-$ increase above the proposed +5 mmol·L$^{-1}$ 'ergogenic threshold' when measured at 60 and 90 min post-ingestion, respectively. Despite similar HCO$_3^-$ increases, however, both of these studies differed in performance outcomes (time to complete 4 x 50 m swimming sprints: 1.3% faster [34] vs. time to complete 6 x 35 m running sprints: no change [35]). Why this discrepancy occurred remains unknown, though the authors speculated that individual factors such as genetics and gastrointestinal (GI) physiology may have had a role [35]. Indeed, both studies did not account for individual differences since standardized ingestion strategies were employed and HCO$_3^-$ changes were reported at the group level only. In order to optimize NaHCO$_3$ supplementation for adolescent athletes, it is therefore necessary to first understand the possible acid-base balance alterations that occur following ingestion. The purpose of this study was to explore the time course changes and peak blood concentrations of three acid-base balance variables (HCO$_3^-$, pH, and the SID) in highly trained adolescent swimmers following the ingestion of 0.3 g·kg BM$^{-1}$ NaHCO$_3$.

## Materials and methods

### Participants

Twelve national level swimmers from a performance swim programme volunteered to participate in this study (Table 1). Three of the swimmers had recently represented Great Britain at either the European Junior Swimming Championships or the European Youth Olympic Festival in 2019, whilst seven had qualified for the senior British Swimming Championships in April 2020. At the time of the study, all swimmers were ranked within the top 25 in Great Britain in at least one event within their respected age group (mean FINA points = 696 ± 62) and were completing a minimum of 7 x 2.5 hours pool-based and 2 x 1-hour land-based training sessions per week. The study was granted institutional ethical approval (Birmingham City

**Table 1. Characteristics of study participants (± SD).**

| Characteristics | Males (n = 5) | Females (n = 7) | Combined (n = 12) |
|---|---|---|---|
| Age (years) | 16.4 ± 1.1 | 15.6 ± 0.8 | 15.9 ± 1.0 |
| Height (m) | 1.78 ± 0.6 | 1.70 ± 0.4 | 1.73 ± 0.6 |
| Body mass (kg) | 72.4 ± 10.7 | 60.3 ± 4.8 | 65.3 ± 9.6 |
| Sum of 8 skinfolds (mm) | 55.2 ± 11.3 | 82.1 ± 21.5 | 70.9 ± 22.1 |
| Time as a competitive junior swimmer (years) | 8.2 ± 1.6 | 8.0 ± 0.8 | 8.1 ± 1.2 |
| Personal best swimming times (freestyle–long course) | | | |
| 50 m (sec) | 26.0 ± 2.6 | 29.4 ± 1.5 | 28.0 ± 2.6 |
| 100 m (sec) | 54.7 ± 3.0 | 61.2 ± 1.4 | 58.8 ± 15.6 |
| 200 m (min: sec) | 1:57.7 ± 4.9 | 2:11.3 ± 3.6 | 2:05.6 ± 8.1 |

University Ethics Committee: Newbury/3649/R(B)/2019 /Nov/HELS FAEC) and both the swimmers and their parents/guardians provided written informed consent prior to their participation in the study.

## Pre-experimental procedures

Each participant was required to attend the laboratory as per their usual training time (17:00–20:30 hrs) having eaten as they normally would prior to a competitive race (i.e., self-selected timing and meal composition). Previous studies have assessed time course changes in $HCO_3^-$ and pH in a fasted state [2, 3, 36] or following a standardized meal [22], however, these conditions do not replicate the variance in nutritional intakes that would take place in competition. This ingestion method was selected to consider the effects of individualized nutrition on baseline acid-base balance, electrolyte status, and absorption rate that would occur in practice [37]. In addition, the ingestion of a meal has shown to reduce the occurrence and severity of potential GI side effects that are associated with $NaHCO_3$ ingestion [38]. Each swimmer completed a 24-hour dietary recall before $NaHCO_3$ ingestion, which was used to calculate diet composition (Table 2) using an online nutrition application (Nutritics, version 5.64, Dublin, Ireland). Similarly, water was permitted to be consumed *ad libitum* (mean intake: 1.2 ± 0.5 L, range: 0.5–2.0 L), though ingestion of further nutrients were restricted once $NaHCO_3$ had been consumed. Participants were also asked to refrain from additional exercise outside of their regular swim training programme in the 48 hours prior to participating in the study. No swimmers had ingested caffeine, although concurrent ingestion of creatine (participants 5, 8, 9, 10, and 11) and beta-alanine (participants 9 and 10) were reported. Neither of these supplements was expected to affect the blood acid-base balance alterations that would occur following $NaHCO_3$ ingestion [39, 40]. All trials were carried out between December 2019 and February 2020 during a specific race preparation training period (weekly swimming volume = 50.9 ± 3.4 km).

## Protocol and measurements

Upon arrival to the laboratory, participants engaged in five min of seated rest before a baseline capillary sample of whole blood was collected from the fingertip into a 100 μL sodium-heparin coated glass clinitube (Radiometer Medical Ltd., Copenhagen, Denmark). Blood samples were immediately analysed using a blood gas analyser (ABL9, Radiometer Medical Ltd., Copenhgen,

**Table 2. The dietary intake of the participant cohort in the 24 hours preceding NaHCO₃ ingestion.**

| | 24 hours pre-ingestion | 1–3 hours pre-ingestion |
|---|---|---|
| Total energy (kcal) | 2852 ± 926 | 655 ± 285 |
| Relative energy (kcal·kg BM⁻¹) | 43.3 ± 9.8 | 10.0 ± 3.7 |
| Total carbohydrate (g) | 362 ± 135 | 75.4 ± 46.0 |
| Relative carbohydrate (g·kg BM⁻¹) | 5.5 ± 1.4 | 1.2 ± 0.6 |
| Total protein (g) | 153 ± 37 | 30.9 ± 9.0 |
| Relative protein (g·kg BM⁻¹) | 2.3 ± 0.4 | 0.5 ± 0.1 |
| Total fat (g) | 88.4 ± 31.1 | 25.6 ± 13.4 |
| Relative fat (g·kg BM⁻¹) | 1.3 ± 0.4 | 0.4 ± 0.2 |
| Total Na⁺ (mg) | 3211 ± 1110 | 901 ± 409 |
| Total K⁺ (mg) | 3546 ± 1185 | 559 ± 560 |
| Total Ca²⁺ (mg) | 1428 ± 501 | 276 ± 229 |
| Total Cl⁻ (mg) | 4755 ± 1558 | 1374 ± 601 |

$Na^+$ = Dietary sodium. $K^+$ = Dietary potassium. $Ca^{2+}$ = Dietary calcium. $Cl^-$ = Dietary chloride.

Denmark) for measurements of $HCO_3^-$, pH, $K^+$, $Na^+$, $Ca^{2+}$, and $Cl^-$. An additional 5 µL sample was taken for the analysis of blood lactate ($La^-$) (Lactate Pro 2, Arkray, Japan) which was used in the following formula to calculate the apparent SID using a freely available spreadsheet: $K^+ + Na^+ + Ca^{2+} - Cl^- - La^-$ [41]. This method of calculating the apparent SID has been previously utilised in similar research investigating the SID alongside $NaHCO_3$ ingestion [24, 27]. Participants then ingested 0.3 g·kg BM$^{-1}$ $NaHCO_3$ (Dr. Oetker, Bielefeld, Germany) contained in gelatine capsules (1 g $NaHCO_3$ per capsule; Size 00, Bulk Powders, Colchester, UK) within a five min period prior to 165 min of quiet seated rest. Blood samples were obtained and analysed on eight more occasions (60, 75, 90, 105, 120, 135, 150, and 165 min post-$NaHCO_3$ ingestion) with further samples taken at 180 (n = 4) and 195 min (n = 1) to ensure a peak $HCO_3^-$ was found. No samples were taken within the first 60 min since 0.3 g·kg$^{-1}$ BM $NaHCO_3$ in capsule form was not expected to display peak $HCO_3^-$ concentrations in this time [3, 22].

Gastrointestinal side effects were monitored using nine 200 mm visual analogue scales (VAS) for nausea, flatulence, stomach cramping, belching, stomach ache, bowel urgency, diarrhoea, vomiting and stomach bloating consistent with previous $NaHCO_3$ research [2, 24, 25, 42]. The VAS scales were labelled "no symptom" on the left side and "severe symptom" on the right side.

## Statistical analysis

Data was normally distributed (Shapiro-Wilks) and sphericity was assumed (Mauchly) prior to statistical analysis. If sphericity was violated, Huyn-Feldt (epsilon value >0.75) or Greenhouse-Geiser (epsilon value <0.75) corrections were applied. A repeated measures ANOVA was conducted to establish mean differences between the time to peak of the three alkalotic variables ($HCO_3^-$, pH, and the SID) and group mean differences for each blood metabolite ($HCO_3^-$, pH, SID, $K^+$, $Na^+$, $Ca^{2+}$, and $Cl^-$) at each sampling time point. Post hoc comparisons were determined by the Bonferroni correction and effect sizes are reported as partial eta squared ($P\eta^2$). Additional effect sizes were calculated for each time course change using the Hedges' $g$ bias correction, which was selected based on the small sample size (n <20) included in this study [43]. Effect sizes were interpreted as trivial ($g$ = <0.20), small ($g$ = 0.20–0.49), medium ($g$ = 0.50–0.79), and large ($g$ = ≥0.80) [44]. Coefficient of variation (CV) was calculated using SD/mean*100 and described the inter-individual variance between participants for time course and absolute blood analyte changes. A Spearman's rank-order correlation ($r_s$) was used to investigate associations between dietary intake (energy, macronutrients, and electrolytes consumed 24 and 3 hours before to $NaHCO_3$ ingestion) and the time to peak, absolute change, and baseline measures of blood $HCO_3^-$, pH, and the SID. This non-parametric correlation was selected due to violations in the sphericity of nutritional data. All statistical tests were completed using Statistical Package for the Social Sciences (SPSS), version 25 (IBM, Chicago, USA). All data are reported as mean ± SD with statistical significance set at p <0.05.

## Results

### Blood analyte changes

No statistical significance was found between the time to peak concentrations of each blood acid-base variable ($HCO_3^-$ = 130 ± 34 min, pH = 120 ± 38 min, SID = 98 ± 37 min; p = 0.077, $P\eta^2$ = 0.208). Despite this, a large effect size ($g$ = 0.88) separated the time to peak SID and $HCO_3^-$ following a mean difference of 33 min being observed between these two measures. A moderate effect size was also observed between time to peak SID and pH (23 min, $g$ = 0.58), whereas the effect size was small between time to peak $HCO_3^-$ and pH (10 min, $g$ = 0.26). Large inter-individual variations were present in the time to peak (CV: $HCO_3^-$ = 27%;

**Table 3. Individual metabolite responses to 0.3 g·kg BM$^{-1}$ sodium bicarbonate (NaHCO$_3$).**

| | Participant (Gender) | | | | | | | | | | | |
|---|---|---|---|---|---|---|---|---|---|---|---|---|
| | 1 (M) | 2 (M) | 3 (F) | 4 (F) | 5 (M) | 6 (M) | 7 (F) | 8 (F) | 9 (F) | 10 (M) | 11 (F) | 12 (F) |
| HCO$_3^-$ baseline (mmol·L$^{-1}$) | 26.5 | 28.0 | 23.3 | 23.9 | 26.0 | 25.6 | 24.8 | 25.9 | 24.7 | 26.4 | 26.4 | 23.6 |
| HCO$_3^-$ peak (mmol·L$^{-1}$) | 34.3 | 35.3 | 32.3 | 34.5 | 32.8 | 33.9 | 31.4 | 28.3 | 30.5 | 32.7 | 34.8 | 26.4 |
| Absolute change (mmol·L$^{-1}$) | +7.8 | +7.3 | +9.0 | +10.6 | +6.8 | +8.3 | +6.6 | +2.4 | +5.8 | +6.3 | +8.4 | +2.8 |
| Time to peak (min) | 90 | 75 | 120 | 150 | 150 | 165 | 75 | 120 | 150 | 150 | 135 | 180 |
| pH baseline (units) | 7.41 | 7.46 | 7.40 | 7.40 | 7.40 | 7.41 | 7.39 | 7.41 | 7.41 | 7.40 | 7.38 | 7.43 |
| pH peak (units) | 7.57 | 7.52 | 7.52 | 7.54 | 7.49 | 7.53 | 7.54 | 7.45 | 7.51 | 7.48 | 7.49 | 7.47 |
| Absolute change (units) | +0.16 | +0.06 | +0.12 | +0.14 | +0.09 | +0.12 | +0.15 | +0.04 | +0.10 | +0.08 | +0.11 | +0.04 |
| Time to peak (min) | 90 | 75 | 90 | 135 | 180 | 165 | 90 | 105 | 165 | 150 | 120 | 75 |
| SID baseline (mEq·L$^{-1}$) | 33 | 37 | 33 | 32 | 33 | 35 | 33 | 33 | 31 | 30 | 34 | 32 |
| SID peak (mEq·L$^{-1}$) | 36 | 38 | 38 | 38 | 37 | 35 | 38 | 35 | 35 | 38 | 38 | 34 |
| Absolute change (mEq·L$^{-1}$) | +3 | +1 | +5 | +6 | +4 | 0 | +5 | +2 | +4 | +8 | +4 | +2 |
| Time to peak (min) | 90 | 60 | 90 | 135 | 120 | 60 | 165 | 120 | 60 | 135 | 75 | 60 |

pH = 32%; SID = 37%;) and absolute increase from baseline observed for each alkalotic measure (HCO$_3^-$ = +6.8 ± 2.4 mmol·L$^{-1}$, CV = 35%; pH = +0.10 ± 0.04 units, CV = 42%; SID = +3.7 ± 2.2 mEq·L$^{-1}$, CV = 61%). Individual data for all blood acid-base balance variables is presented in Table 3, whereas the time course changes in HCO$_3^-$ and the SID has been plotted in Figs 1 and 2 for four different participants to emphasise the individuality in blood responses.

Across the sampling timeframe, marked increases in acid-base variables were observed (HCO$_3^-$: p <0.001, Pη$^2$ = 0.609; pH: p <0.001, Pη$^2$ = 0.529; SID: p = 0.029, Pη$^2$ = 0.212; Fig 3). Blood bicarbonate was elevated 60 min post-ingestion (+3.2 ± 2.6 mmol·L$^{-1}$, p = 0.046) and remained +4–6 mmol·L$^{-1}$ until 165 min post-ingestion (all p <0.030). At 105 min mean blood HCO$_3^-$ had increased by 5.2 ± 2.5 mmol·L$^{-1}$ and remained elevated above the proposed

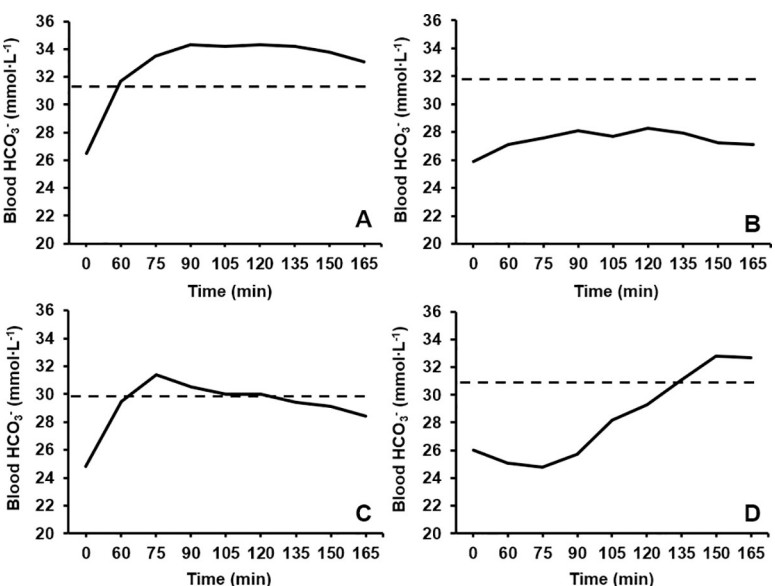

**Fig 1.** The time course change in blood bicarbonate (HCO$_3^-$) over a 165 min period following NaHCO$_3$ ingestion for A) participant 1, B) participant 8, C) participant 7, and D) participant 5. The dotted line represents a +5 mmol·L$^{-1}$ increase from baseline blood HCO$_3^-$.

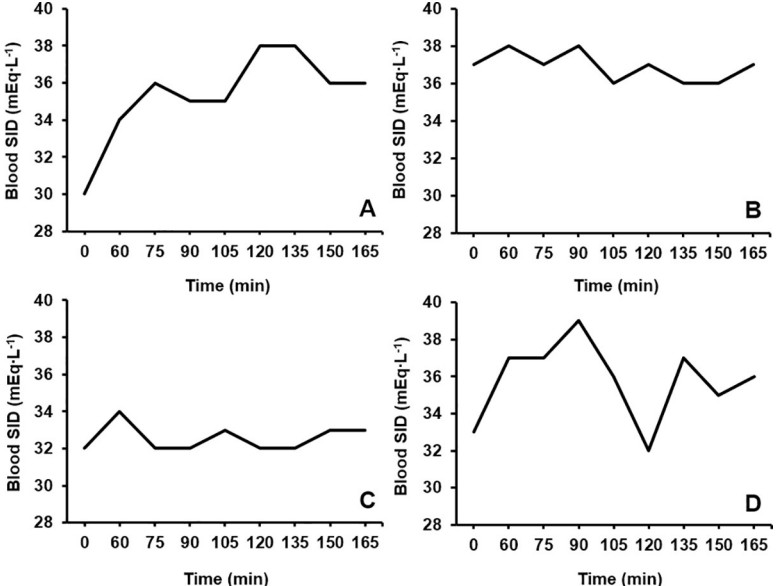

**Fig 2.** The time course change in blood strong ion difference (SID) over a 165 min period following NaHCO₃ ingestion for A) participant 10, B) participant 2, C) participant 12, and D) participant 3.

+5 mmol·L⁻¹ ergogenic threshold until the final 165 min time point (peak: 150 min, +5.9 ± 2.7 mmol·L⁻¹, +23%). An increased pH occurred at 75 min post-ingestion (+0.06 ± 0.04 units, +0.8%, p = 0.010) and this level of increase was sustained at all remaining points in time (+0.06–0.08 units, all p <0.030). Peak pH occurred at 165 min post-ingestion (+0.08 ± 0.05 units, +1.1%). The SID increased by 2.0 ± 1.5 mEq·L⁻¹ (+6%, p = 0.024) at 60 min post-inges-tion and peaked at 135 min post-ingestion (+2.4 ± 2.6 mEq·L⁻¹, +7%), however, this did not reach statistical significance.

Ingestion of NaHCO₃ produced ionic shifts in the balance of strong ions (K⁺: p = 0.005, Pη² = 0.215; Na⁺: p = 0.044, Pη² = 0.160; Ca²⁺: p <0.001, Pη² = 0.562; Cl⁻: p = 0.016, Pη² = 0.306). The electrolytes with the greatest absolute changes were K⁺ and Cl⁻. Large inter-indi-vidual variations restricted the ability to detect statistical significances for all metabolites, therefore group level results are described using effect sizes (Fig 4). This analysis showed that K⁺ reduced gradually from baseline before reaching peak declines at 120 (–0.93 ± 1.07 mmol·L⁻¹, –18%, g = 1.26) and 150 min (–1.03 ± 0.93 mmol·L⁻¹, –20%, g = 1.64) post-ingestion. Large effect sizes were calculated for the reductions in Cl⁻ from 105 to 165 min post-ingestion (all g >0.80) with the lowest point also occurring at 150 min post-ingestion (–3.5 ± 1.9 mmol·L⁻¹, –3%, p = 0.002). The time that each individual participant reached their lowest val-ues of K⁺ (133 ± 28 min, CV = 21%), Cl⁻ (124 ± 27 min, CV = 21%), and Ca²⁺ (141 ± 32 min, CV = 23%) averaged within 20 min of one another, albeit with large inter-individual variances. Conversely, Na⁺ was largely unchanged throughout the sampling period. The absolute changes from baseline displayed high inter-individual variation for each variable (K⁺: –1.28 ± 0.88 mmol·L⁻¹, CV = 69%; Cl⁻: –4.5 ± 2.1 mmol·L⁻¹, CV = 47%; Ca²⁺: –0.16 ± 0.07 mmol·L⁻¹, CV = 43%; Na⁺: +1.6 ± 1.3 mmol·L⁻¹, CV = 83%).

The energy and macronutrient composition of the pre-ingestion meal had no statistically significant correlations with the increases observed in blood HCO₃⁻, pH, or the SID (Table 4). However, the macronutrient consumption of the meal did affect the time course behaviour of blood HCO₃⁻, with protein (rs = –0.581) and fat intakes (rs = –0.594) both having moderate, negative correlations with time to peak HCO₃⁻. Conversely, protein intake had a strong,

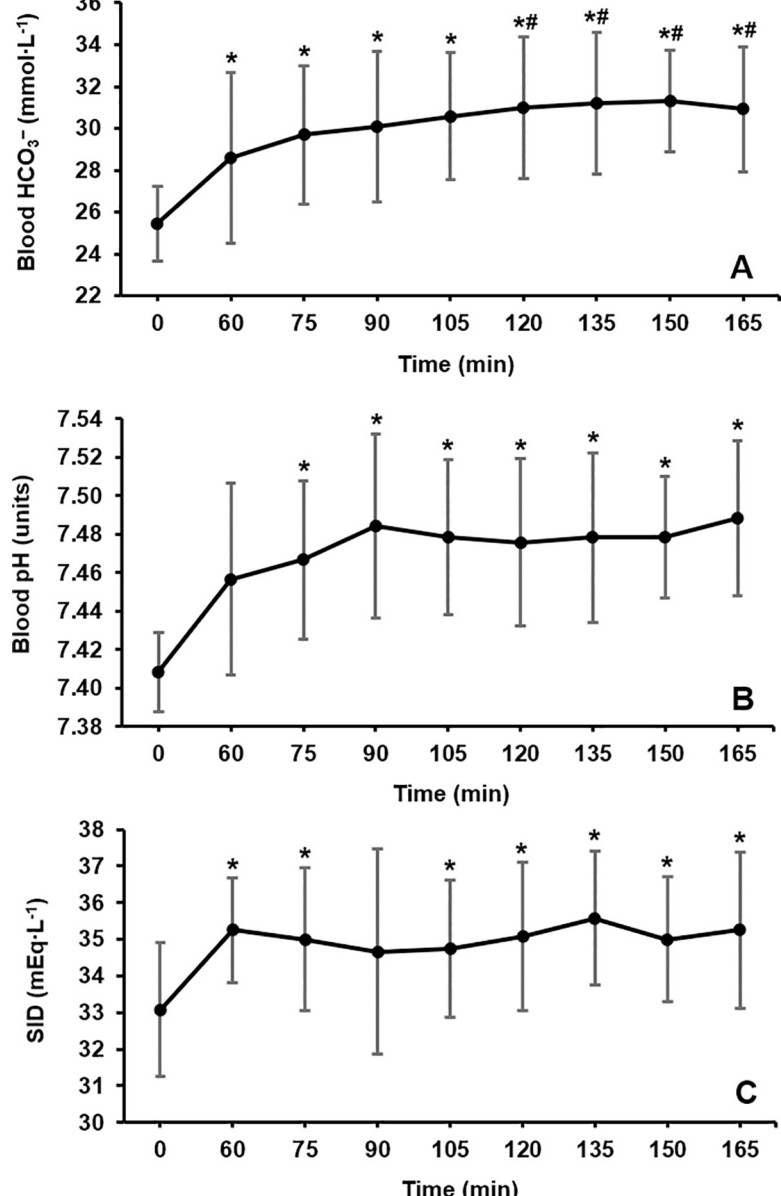

**Fig 3.** Group mean time course changes in A) blood bicarbonate (HCO$_3^-$), B) pH, and C) the strong ion difference (SID) over a 165 min period following NaHCO$_3$ ingestion. $^*$ = a large effect size compared to baseline ($g$ >0.80). $^\#$ = a large effect size compared to 60 min post-ingestion ($g$ >0.80).

positive correlation ($r_s$ = 0.711) with time to peak SID. The total energy content of the pre-ingestion also had a moderate, positive correlation with time to peak SID, although this did not reach statistical significance ($r_s$ = 0.555, p = 0.061). The fat content of the pre-ingestion meal was associated with baseline blood acid-base balance, as both HCO$_3^-$ ($r_s$ = 0.609) and the SID ($r_s$ = 0.639) displayed strong, positive correlations. The amount of Na$^+$ consumed 1–3 hours before NaHCO$_3$ ingestion also had a strong, positive correlation with baseline HCO$_3^-$ ($r_s$ = 0.644, p = 0.024), but not with pH ($r_s$ = –0.315, p = 0.318), or the SID ($r_s$ = 0.171, p = 0.596). No other correlations were found between Na$^+$, K$^+$, Cl$^-$, or Ca$^{2+}$ ingestion and time course changes in blood acid-base balance.

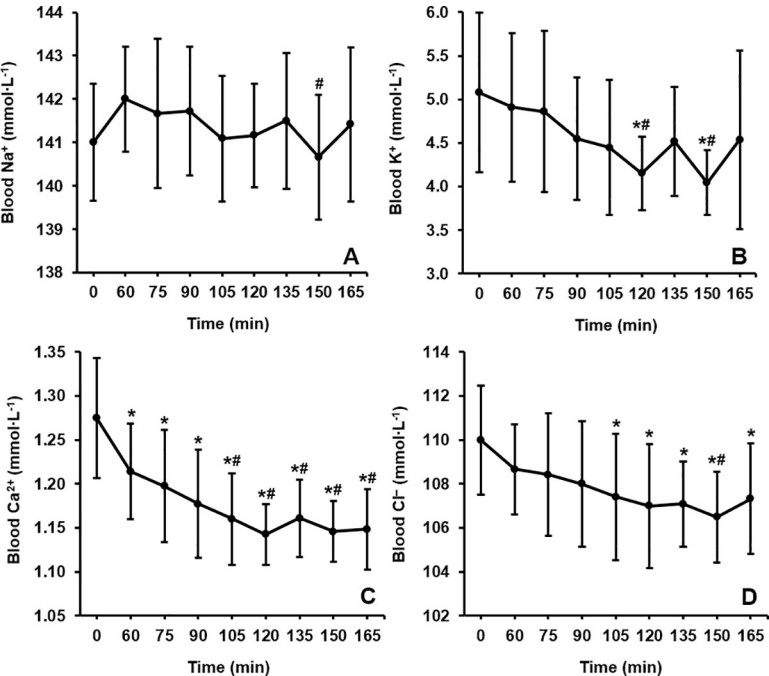

**Fig 4.** Group mean time course changes in A) blood sodium ($K^+$), B) potassium ($Na^+$), C) calcium ($Ca^{2+}$), and D) chloride ($Cl^-$) over a 165 min period following $NaHCO_3$ ingestion. * = a large effect size compared to baseline ($g$ >0.80). # = a large effect size compared to 60 min post-ingestion ($g$ >0.80).

## Gastrointestinal discomfort

All participants reported at least one GI side effect, though the severity of these instances was rated ≤5/10 in 94% (29/31) of occurrences (Table 5). Of the reported side effects, 71% (22/31)

**Table 4. The associations between the composition of the pre-ingestion meal with the time course changes in acid-base balance variables following NaHCO₃ ingestion.**

| Dietary intake 1–3 hours before NaHCO₃ ingestion | Correlations with time to peak | | |
|---|---|---|---|
| | HCO₃⁻ (min) | pH (min) | SID (min) |
| Energy (kcal) | p = 0.202, $r_s$ = −0.397 | p = 0.785, $r_s$ = −0.088 | p = 0.061, $r_s$ = 0.555 |
| Carbohydrate (g) | p = 0.506, $r_s$ = −0.213 | p = 0.784, $r_s$ = −0.088 | p = 0.194, $r_s$ = 0.403 |
| Protein (g) | **p = 0.048, $r_s$ = −0.581*** | p = 0.870, $r_s$ = −0.053 | **p = 0.010, $r_s$ = 0.711*** |
| Fat (g) | **p = 0.042, $r_s$ = −0.594*** | p = 0.411, $r_s$ = −0.262 | p = 0.539, $r_s$ = 0.197 |
| Dietary intake 1–3 hours before NaHCO₃ ingestion | Correlations with absolute change | | |
| | HCO₃⁻ (mmol·L⁻¹) | pH | SID (mEq·L⁻¹) |
| Energy (kcal) | p = 0.430, $r_s$ = 0.252 | p = 0.084, $r_s$ = 0.519 | p = 0.827, $r_s$ = 0.071 |
| Carbohydrate (g) | p = 0.372, $r_s$ = 0.284 | p = 0.127, $r_s$ = 0.466 | p = 0.814, $r_s$ = 0.076 |
| Protein (g) | p = 0.476, $r_s$ = 0.228 | p = 0.054, $r_s$ = 0.569 | p = 0.221, $r_s$ = 0.381 |
| Fat (g) | p = 0.846, $r_s$ = 0.063 | p = 0.316, $r_s$ = 0.316 | p = 0.259, $r_s$ = −0.354 |
| Dietary intake 1–3 hours before NaHCO₃ ingestion | Correlations with baseline concentrations | | |
| | HCO₃⁻ (mmol·L⁻¹) | pH | SID (mEq·L⁻¹) |
| Energy (kcal) | p = 0.285, $r_s$ = 0.336 | p = 0.645, $r_s$ = −0.149 | p = 0.569, $r_s$ = 0.171 |
| Carbohydrate (g) | p = 0.652, $r_s$ = 0.146 | p = 0.801, $r_s$ = −0.082 | p = 0.920, $r_s$ = 0.033 |
| Protein (g) | p = 0.724, $r_s$ = 0.114 | p = 0.114, $r_s$ = −0.480 | p = 0.826, $r_s$ = 0.071 |
| Fat (g) | **p = 0.036, $r_s$ = 0.609*** | p = 1.000, $r_s$ = <0.001 | **p = 0.025, $r_s$ = 0.639*** |

**Table 5. The three most severe symptoms of gastrointestinal discomfort experienced following the ingestion of 0.3 g·kg BM$^{-1}$ sodium bicarbonate (NaHCO$_3$).**

| Participant (Body mass) | Caps | Symptom 1 | Severity (n/10) | Peak (mins) | Symptom 2 | Severity (n/10) | Peak (mins) | Symptom 3 | Severity (n/10) | Peak (mins) | Aggregated |
|---|---|---|---|---|---|---|---|---|---|---|---|
| 1 (90.7 kg) | 28 | Belching | 4.9 | 60 | Stomach bloating | 4.9 | 60 | Nausea | 2.3 | 60 | 17.7 |
| 2 (69.5 kg) | 21 | Belching | 2.7 | 60 | Stomach bloating | 1.7 | 135 | Stomach cramp | 1.1 | 60 | 6.6 |
| 3 (55.8 kg) | 17 | Nausea | 4.1 | 75 | Stomach bloating | 1.4 | 60 | None | 0.0 | N/A | 5.5 |
| 4 (62.2 kg) | 19 | Belching | 3.9 | 60 | Stomach bloating | 2.4 | 60 | Nausea | 0.7 | 75 | 6.9 |
| 5 (72.1 kg) | 22 | Belching | 1.7 | 75 | None | 0.0 | N/A | None | 0.0 | N/A | 1.7 |
| 6 (64.6 kg) | 20 | Stomach ache | 4.0 | 165 | None | 0.0 | N/A | None | 0.0 | N/A | 4.0 |
| 7 (60.6 kg) | 19 | Nausea | 1.4 | 60 | None | 0.0 | N/A | None | 0.0 | N/A | 1.4 |
| 8 (54.3 kg) | 17 | Belching | 4.9 | 60 | Flatulence | 3.9 | 165 | Stomach ache | 1.5 | 135 | 11.0 |
| 9 (57.7 kg) | 18 | Nausea | 2.6 | 60 | None | 0.0 | N/A | None | 0.0 | N/A | 2.6 |
| 10 (65.0 kg) | 20 | Belching | 5.0 | 60 | None | 0.0 | N/A | None | 0.0 | N/A | 5.0 |
| 11 (67.9 kg) | 21 | Flatulence | 2.6 | 150 | Stomach bloating | 2.1 | 60 | Bowel urgency | 1.5 | 165 | 6.1 |
| 12 (63.7 kg) | 20 | Vomiting | 10.0 | 105 | Nausea | 10.0 | 105 | Stomach ache | 1.5 | 90 | 22.3 |

Aggregated = Sum of the most severe incidence of all nine symptoms (n/90). Caps = Number of capsules consumed (1 g of NaHCO$_3$ per capsule).

of instances peaked in severity at 60 and 75 min post-ingestion. The most common symptom was belching (58%, 7/12), followed by stomach bloating and nausea (50%, 6/12).

## Discussion

This was the first study to compare the time to peak of three different methods of measuring peak alkalosis (i.e., $HCO_3^-$, pH, and the SID) following the ingestion of 0.3 g·kg BM$^{-1}$ NaHCO$_3$ in highly trained adolescent male and female swimmers. Though no statistical significance was observed between the three measures, there was a 33 min difference between the average time that $HCO_3^-$ and the SID peaked in the blood of trained adolescents. This study also identified large inter-individual variances in the time to peak of each alkalotic measure, whereby individual peak concentrations were identified across the sampling timeframe (between 60–180 min post-ingestion). It is possible that this difference was related to the protein content of a meal consumed 1–3 hours before NaHCO$_3$ ingestion, with this macronutrient positively and negatively correlated with time to peak SID and $HCO_3^-$, respectively. Finally, this study provides justification for utilising individualised NaHCO$_3$ dosing strategies in adolescents since some sustained $HCO_3^-$ increases exceeding 5 mmol·L$^{-1}$ for over 60 min (n = 6), whereas others either reached this threshold for a 30–60 min period (n = 4), or did not reach this threshold at all (n = 2). Based on the individuality in blood responses, future work should aim to clarify whether personalised NaHCO$_3$ strategies (i.e., time to peak $HCO_3^-$ and time to peak SID) have a greater ergogenic effect for adolescents than standardised dosing procedures.

The average time that $HCO_3^-$ and pH peaked within the blood of trained adolescents occurred within 10 min of one another, supporting the relationship between these two measurements when NaHCO$_3$ is consumed in gelatine capsules [3, 22, 38]. This discussion will therefore focus on the blood $HCO_3^-$ time course since this measure displays less variance and

greater reliability compared to pH [2, 21]. Time to peak blood HCO$_3^-$ varied considerably within trained adolescents (75–180 min), which is in accordance with the blood responses of recreational adults following the ingestion of NaHCO$_3$ capsules in a fed or fasted state [3, 21, 22]. Whereas no comparisons can presently be made for the time course changes in the SID, the time to peak concentration was also highly variable between each individual (60–165 min). Interestingly, the average time that the SID peaked in trained adolescents occurred 33 min before time to peak HCO$_3^-$, which may have implications for NaHCO$_3$ timing in applied practice. Why this time difference occurred is currently unknown, though meal preference in the 1–3 hours prior to NaHCO$_3$ ingestion may be a contributing factor. Indeed, higher protein intakes were correlated ($r_s$ = 0.711) with a delayed time to peak SID, whereas in contrast, higher protein intakes were also correlated ($r_s$ = –0.581) with a faster time to peak blood HCO$_3^-$. Since NaHCO$_3$ will usually be supplemented with a pre-competition meal in practice, it is plausible that these two measurements will also be separated prior to performance. However, since this research did not measure a performance outcome, future work should aim to determine whether either of these individualised approaches (i.e., time to peak SID and HCO$_3^-$) has an enhanced ergogenic benefit versus a traditional, fixed ingestion method (i.e., 120 min pre-exercise).

The participants in the present study demonstrated a peak blood HCO$_3^-$ concentration that exceeded the 5–6 mmol·L$^{-1}$ threshold (+6.8 ± 2.4 mmol·L$^{-1}$), contrasting previous studies when trained adolescents ingested NaHCO$_3$ prior to exercise [34, 35]. A possible explanation for this difference is that high-level swimmers often have an early maturation onset [45], which likely occurred in the current participants and aided their selection for the high-performance swimming programme in which they were recruited. For comparison, both the male (72 ± 11 kg) and female (60 ± 5 kg) swimmers from the present study carried considerably more body mass than the 'well trained' male swimmers (56 ± 1 kg) recruited by Zajac et al. [34], despite being the same age. Consequently, the current participants may have benefitted from maturity-related differences in gut physiology (i.e., caecal pH, distal microbiota) that potentially enhanced their capacity to uptake NaHCO$_3$ from the GI tract [46, 47]. Moreover, the advanced training age of the participants could have also enhanced muscle and blood buffering capabilities (e.g., monocarboxylate transporter density, Na$^+$/HCO$_3^-$ co-transport) [48–50], which could have enabled greater blood HCO$_3^-$ increases to occur. However, it should be noted that two participants (both aged 15 years) did not increase their blood HCO$_3^-$ concentrations above 3 mmol·L$^{-1}$, therefore adding support to previous speculations that less biologically mature adolescents might have less capacity increase blood HCO$_3^-$ by more than 5–6 mmol·L$^{-1}$. Though it is unclear whether this range is necessary for performance enhancement in adolescents, it is perhaps appropriate to suggest that less biologically mature individuals do not require NaHCO$_3$ supplementation at their stage of development [51].

Highly trained adolescents also demonstrated an increased blood SID following NaHCO$_3$ ingestion. This supports contemporary evidence by Gough et al. [24, 27] whereby an elevated SID was reported prior to exercise in trained adults. Surprisingly, both studies by Gough et al. demonstrated greater group mean increases in the SID compared to the present study (+4 vs. +6–10 mEq·L$^{-1}$), despite not purposely seeking to identify a peak concentration. The current study also observed lower baseline (33 vs. 36–38 mEq·L$^{-1}$) and absolute peak SID concentrations (37 vs. 42–46 mEq·L$^{-1}$), inferring that adolescents might also be limited with their capacity for strong ion movements (e.g., reduced Na$^+$/K$^+$ pump activity) [48]. Furthermore, these lower concentrations occurred despite food consumption 1–3 hours before NaHCO$_3$ ingestion, though the energy, macronutrient, or electrolyte (Na$^+$, K$^+$, Cl$^-$, and Ca$^{2+}$) composition of this meal was not correlated with baseline SID or the absolute SID increases that were achieved. Since water was also permitted to be consumed *ad libitum*, it is possible that plasma

volume also increased and masked some of the ionic movements that took place [52]. For example, the absolute increases observed in plasma $Na^+$ were mitigated (+1.6 vs. +3.0 mmol·$L^{-1}$) in comparison to previous research that investigated $NaHCO_3$ ingestion in a fasted state [2, 3, 27]. Nonetheless, large effect sizes were calculated for plasma reductions of $K^+$, $Ca^{2+}$, and $Cl^-$ between 120–150 min post-ingestion, therefore suggesting that an intramuscular uptake of these ions did occur at this time [53]. Given that this study did not consider changes in the intramuscular SID or plasma volume, however, future research is required in order to further elucidate the role of $NaHCO_3$ on the SID during exercise performance.

Comparable to adults, each blood analyte displayed a large inter-individual variance with regards to time to peak and absolute increases that occurred from baseline concentrations [2, 3, 22]. Why this phenomenon occurs remains elusive, though it is postulated that genetic differences in the rate of gastric emptying, intestinal motility, and GI blood flow may contribute to $NaHCO_3$ uptake characteristics [54, 55]. Despite the individual differences, the process of identifying a personal time to peak $HCO_3^-$ measurement has recently been criticised [22]. Ingesting $NaHCO_3$ capsules has consistently been shown to produce increases in blood $HCO_3^-$ (>5 mmol·$L^{-1}$) that last for over 100 min in adults [3, 22, 23], therefore producing a long lasting 'ergogenic window' that undermines the necessity of a single ingestion time point. However, when considering the $HCO_3^-$ time course on an individual basis, only half (n = 6) of the trained adolescent participants demonstrated a +5 mmol·$L^{-1}$ $HCO_3^-$ increase that lasted for more than 90 min. Moreover, the time that each participant spent above the +5 mmol·$L^{-1}$ threshold occurred at completely separate periods within some individuals (i.e., 60–120 min, 135–165 min post-ingestion), whereas two participants had small reductions in blood $HCO_3^-$ (~1 mmol·$L^{-1}$) at 60–75 min post-ingestion before concentrations increased over the next 60–75 min. Due to these erratic time course differences, it is recommended that adolescents undergo time to peak testing prior to supplementation in order to identify a personal time frame for $NaHCO_3$ to elicit a sufficient blood $HCO_3^-$ response. However, considering that this is an expensive procedure, adolescents should remain cautious since the effects of individualised ingestion methods on exercise performance are currently untested within this cohort.

There were minimal incidents of severe GI side effects when trained adolescents ingested $NaHCO_3$ capsules 1–3 hours after consuming a self-selected meal. Whilst each participant reported at least one side effect, the severity of these symptoms was rated less than 5 out of 10 on 94% (29/31) of occasions. This low level of GI disturbance therefore supports the findings of Carr et al. [38], whereby the co-ingestion of $NaHCO_3$ capsules with a meal provided the best mitigation against GI distress (vs. $NaHCO_3$ solution/fasted state). Both capsules and meal consumption are suggested to slow the reduction of $NaHCO_3$ to $Na^+$ and carbonic acid ($H_2CO_3$), subsequently reducing the rate of acid-base disturbances that occur in the stomach (e.g., increases in carbon dioxide and water) [56, 57]. Alternatively, the age of the participants might have contributed to the lack of GI symptoms observed in this study, therefore supporting the lack of GI disturbances in adolescents from previous $NaHCO_3$ research [34, 35]. Whilst there is currently no clear explanation for the lack of GI disturbances in adolescents, it is plausible that a lower acidity of the GI fluids compared to adults may mediate the acid-base disturbances that occur in the stomach when an alkaline substance is ingested [58]. Another general observation is that athletes of a higher body mass (>75 kg) typically display greater GI distress than lighter individuals [42, 59]. Adolescents, being smaller in stature than their adult peers, may therefore experience less GI disturbances due to a reduced absolute $NaHCO_3$ dosage entering the stomach. However, this study did observe vomiting in one participant, which indicates that the severity GI side effects should be assessed on an individual basis.

The study methods were applied based on the logistical (e.g., taking highly trained swimmers out of training) and ethical considerations of the participant cohort (e.g., repeated blood

analysis of young athletes). This limited the current study from including a placebo experiment or repeatability measures that could have altered the interpretation of results. Nonetheless, previous work found a placebo to cause no blood time course changes for either HCO$_3^-$, pH, or Na$^+$ [2, 22], therefore it may be assumed that a placebo would not have altered baseline HCO$_3^-$, pH, or SID within an adolescent cohort. The repeatability of time to peak HCO$_3^-$ is controversial, however, when 0.3 g·kg BM$^{-1}$ NaHCO$_3$ capsules are ingested in a fed state. Indeed, Boegman et al. [21] reported an excellent repeatability (ICC = 0.77) within world-class rowers when NaHCO$_3$ was ingested alongside a standardised snack (energy: 682 kcal; protein: 20 g; carbohydrate: 140 g). Conversely, Oliveira et al. [22] found repeatability to be poor (ICC = 0.34) when recreational adults ingested NaHCO$_3$ 60 min following a standardised breakfast (energy: 563 kcal; protein: 9 g; carbohydrate: 90 g). Since both studies differed in meal composition, participant training status, time post-prandial, and blood analysis methods (capillary [21] vs. venous [22] samples), this leaves the repeatability of the current data unclear. However, the main purpose of this research was to identify the time course changes in three alternate blood acid-base balance variables following NaHCO$_3$ ingestion in a highly trained adolescent cohort. Since this was the first study to suggest that a difference in time to peak HCO$_3^-$ and SID may exist in practice, future studies should therefore aim to determine the repeatability of these two measures under different competitive scenarios (e.g., pre-competition meal, after warm-up).

## Conclusion

This study shows that highly trained adolescent swimmers have significant increases and highly individual time course changes in blood HCO$_3^-$, pH, and SID following the ingestion of NaHCO$_3$ capsules. Importantly, the post-ingestion time point in which the individual time to peak occurred for HCO$_3^-$ and the SID was separated by a large effect size. Given that the effects of acidosis on exercise performance are controversial, future studies should seek to identify NaHCO$_3$ dosing strategies based upon a peak SID concentration could be more ergogenic than standardised ingestion strategies, or individualised approaches based upon a time to peak blood HCO$_3^-$. This study also suggests that highly trained adolescent athletes do not have a long lasting 'ergogenic window' following the ingestion of NaHCO$_3$ capsules, therefore supporting the practice of identifying an individualised time point whereby a peak alkalosis occurs. It is acknowledged that a lack of control surrounding pre-ingestion food and fluids may have influenced the findings of this study, however, these conditions better replicate the actual changes that would occur when athletes are in full competition preparation.

## Supporting information

**S1 Data.**
(XLSX)

## Acknowledgments

We gratefully acknowledge all participants who volunteered for the study and the City of Birmingham Swimming Club for facilitating the research process.

## Author Contributions

**Conceptualization:** Lars R. McNaughton.

**Data curation:** Josh W. Newbury.

**Formal analysis:** Josh W. Newbury.

**Funding acquisition:** Lewis A. Gough.

**Methodology:** Josh W. Newbury, Lewis A. Gough.

**Project administration:** Adam L. Kelly, Richard J. Chessor, Lewis A. Gough.

**Supervision:** Richard J. Chessor, S. Andy Sparks, Lars R. McNaughton, Lewis A. Gough.

**Writing – original draft:** Josh W. Newbury, Richard J. Chessor, Lewis A. Gough.

**Writing – review & editing:** Josh W. Newbury, Matthew Cole, Adam L. Kelly, Richard J. Chessor, S. Andy Sparks, Lars R. McNaughton, Lewis A. Gough.

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
