## [Decision Letter · Decision Letter 0]

9 Apr 2021

PONE-D-21-06382

The time to peak blood bicarbonate (HCO3–), pH, and strong ion difference (SID) following sodium bicarbonate (NaHCO3) ingestion in highly trained adolescent swimmers

PLOS ONE

Dear Dr. Sparks,

Thank you for submitting your manuscript to PLOS ONE. After careful consideration, we feel that it has merit but does not fully meet PLOS ONE’s publication criteria as it currently stands. Therefore, we invite you to submit a revised version of the manuscript that addresses the points raised during the review process.

Please address the reviewers comments and respond in a point by point manner.

We look forward to receiving your revised manuscript.

Kind regards,

Caroline Sunderland

Academic Editor

PLOS ONE

Journal Requirements:

'The authors have declared that no competing interests exist.'

We note that one or more of the authors are employed by a commercial company: British Swimming.

Please respond by return email with an updated Funding Statement and Competing Interests Statement and we will change the online submission form on your behalf.

3. Please upload a copy of Figure 3, to which you refer in your text on page 11. If the figure is no longer to be included as part of the submission please remove all reference to it within the text.

Additional Editor Comments (if provided):

Reviewers' comments:

Reviewer's Responses to Questions

**Comments to the Author**

1. Is the manuscript technically sound, and do the data support the conclusions?

Reviewer #1: Partly

Reviewer #2: Partly

2. Has the statistical analysis been performed appropriately and rigorously? 

Reviewer #1: Yes

Reviewer #2: Yes

3. Have the authors made all data underlying the findings in their manuscript fully available?

Reviewer #1: No

Reviewer #2: Yes

4. Is the manuscript presented in an intelligible fashion and written in standard English?

Reviewer #1: Yes

Reviewer #2: Yes

5. Review Comments to the Author

Reviewer #1: The authors have done a nice job in determining the difference in time to peak pH vs. HCO3 vs. SID following ingestion of sodium bicarbonate in a group of adolescent swimmers. The methods are generally sound although I have some concerns about the interpretation and the extrapolation of these results. I hope the authors will consider my suggestions.

The rationale for using adolescents is rather weak. It feels like the authors are trying to justify using this population for a reason that may not have been the initial reason for using this population. Perhaps it was just a convenience sample? I may be wrong of course, but to me it currently appears this way. Likewise, there is no issue with that in my opinion, but it is best to be clear upfront if this was the case.

The limitation of no performance measure should also be highlighted here as the authors focus somewhat on the suggestion that acidosis may not be limiting to performance, but do not investigate this in any way.

I believe the authors must temper statements, extrapolations and conclusions to the sample population used here (adolescents) since this may at least in part explain some of the differences shown here compared to other literature. Specific comments are below.

Line 42: In my opinion, the word “adequate” here undersells the strength of evidence regarding SB supplementation and performance. [See line 56 where the authors themselves state “strong ergogenic benefit”]

Line 65: Interestingly, 60 min post-supplementation in the data of Oliveira et al. showed only a 69% likelihood of having increased >5 mmol/L of HCO3, perhaps supporting these data?

Line 76-77: “A concomitant increase in muscular Cl– uptake is also observed [27,29,30]” – Following SB supplementation? Please clarify.

Line 97-98: This statement is incorrect. Please see https://doi.org/10.1123/ijsnem.2020-0031 as at least one example. Interestingly, this study also showed low bicarbonate increases at a standardised timepoint in adolescents (albeit 90 min post-supplementation). This study does somewhat try to report some individual variation which might help the authors here.

Line 106: “adult-like NaHCO3 absorption” sounds very odd. I suggest rewording.

Lines 114-119: While this is interesting information and the sample is clearly top-class, is this necessary here since there are no performance measures? Does being an athlete change the variables measured here?

178-181 (and 144-146): Did the authors also perform this analysis for the group as a whole? Why would the authors believe creatine and beta-alanine supplementation would influence the responses measured here? Certainly, in my experience working with both supplements, they do not affect any of these measures at rest.

Line 181-182: “Additional effect sizes were calculated where appropriate using Hedges’ g bias correction [41].” – Could the authors please specify exactly where and why it was considered appropriate.

Line 183-184: Coefficient of variation of what? Please specify here.

Line 189-190: Despite no statistical significance, mean differences of as much as 35 min are fairly large in my opinion. It will be important to discuss this in my opinion.

Line 245-247: This is very surprising considering my previous comment. Again, I would like to know the author’s rationale for separating these two groups. Was this decision a priori?

Line 269-271: How many individuals showed a rapid decrease following peak HCO3? And what are the authors considering a rapid decrease here?

Line 271-273: Certainly, in the adolescent study here this might be the case. Please restrict these claims to the population employed.

Line 287-288: Is the bicarbonate absorbed?

Line 289: Although technically correct that creatine could increase buffering, its actual contribution to buffering capacity is extremely low. I suggest not calling them both buffering supplements.

Line 288-291: I would be intrigued to know the authors speculation as to why this occurred. Previous work with beta-alanine and sodium bicarbonate supplementation (https://pubmed.ncbi.nlm.nih.gov/21407127/) has not shown beta-alanine to blunt bicarbonate increased with sodium bicarbonate. I cannot think of a mechanism via which beta-alanine supplementation would blunt these changes. Similarly, I am unaware of why creatine supplementation would either. I think it is important to provide some physiological and mechanistic reasoning and speculation as to why this difference occurred.

Line 302-304: I believe there may be, as mentioned previously [Comment Line 97-98].

Line 315-316: The authors cannot state this since they made no comparison to not ingesting a meal. Please modify this statement, perhaps indicating simply that incidence and intensity of side-effects was low.

Line 317: What is the definition of “minor” here?

Line 317-319: Was the food intake prior to supplement ingestion recorded? This could provide useful and important information. Particularly when comparing the two groups separated according to creatine and beta-alanine co-supplementation.

Line 319: What are the authors referring to here with “This”? The meal in general or the fact it was a high CHO meal?

Line 338-339: “Stark difference” to what? Between sessions? To data from reference 21?

Line 338-340: Could the authors please elaborate why they believe venous vs. capillary blood samples might lead to different time course responses. Likewise, why would the different time frames (180 vs 240 min) affect the repeatability of the two studies?

Line 338: This should be changed to “one hour following a standardised meal” since they were not ingested together.

Line 351-357: I believe some context needs to be provided. Could the absolute dose provided have contributed somewhat? Mean 20 g (0.3 * 65.3 kg) vs 23 g in reference 22? The authors themselves also speculate that co-ingestion of supplements contributed somewhat to the pharmacokinetics. Additionally, these data here are in adolescents and not in adults which might also have modified the results. Thus, some of these statements should be tempered and contextualised. In fact, there is very similar data here to that shown in reference 22 (of which I am a co-author). However, I believe conclusions here should be restricted to the adolescent athlete.

It is also important to consider that the study of Oliveira performed robust statistical modelling of the data, providing statistical likelihoods and probabilities that bicarbonate increases were above +5 and +6 thresholds. This is entirely different to visually stating some people do not always stay above this threshold for prolonged periods. Again, I urge some tempering of statements because the analyses here were entirely different.

Line 356: Since there are some differences in opinion between groups, I would suggest the authors make it clear that it is they specifically making this recommendation here, and not the entire scientific community (e.g. Due to the findings of the current study, we recommend that the individual time point of peak alkalosis…”). Certainly, this intriguing question cannot be conclusively answered either way right now until more work is performed.

Line 369-371: “Given that the effects of acidosis on exercise performance are controversial, this finding suggests that using a time to peak SID approach could be a more appropriate NaHCO3 ingestion strategy in practice.” – I don’t believe that these data suggest this at all since you did not measure exercise performance and compare peak SID to peak HCO3. I suggest this statement be removed.

Line 371-372: This is worded rather oddly (“has an any further ergogenic benefits on exercise performance”). Please consider rephrasing. Likewise, further ergogenic benefit compared to what? Please specify.

Line 373-374: I suggest the authors make it clearer and isolate this only to the adolescent group studied here.

Figure 1. The quality is quite poor and it is difficult to make out the individual timelines (though this may just be due to the uploading process on the journal platform). Perhaps it is just the quality of the figure as it currently stands, but I am struggling to make out many individuals who reached peak HCO3 increases followed by quick decreases (Line 269-271). Could the authors identify individuals consuming other supplements and those not?

Reviewer #2: Newbury et al. rigorously investigated the pharmacokinetics of the increase in strong ions following ingestion of 0.3 g/kg BW sodium bicarbonate in highly trained swimmers. This research group has performed some innovative research on this topic during the last years and is considered as world-leading in the domain of sodium bicarbonate ingestion to improve exercise performance. The authors should be congratulated for using a highly ecological valid design making the study applicable for athletes. My primary concerns are related to the power of the study and to the validity of the measurements obtained.

• L130: The authors try to perform the study as realistic as possible. However, all measurements are performed in a rested state. It would be interesting to investigate the impact of a warming-up on the pharmacokinetics of the strong ion increase.

• L135-137: Did the authors perform some correlation analyses to assess whether the macronutrient composition of the pre-exercise meal is related to the pharmacokinetics of the increase in SID?

• L171: A lot of results were almost significant (p values between 0.05 and 0.10) and effect sizes were quite large. Did the authors perform an a-priori power analysis? I am afraid that a lot of statistical effects failed to reach significance as the power of the study was too low?

• L178: Why did the authors perform a paired t-test and not an unpaired t-test when comparing participants that co-ingested other supplements and those who did not?

• L198 – figure 1. In the legend of figure 1 is indicated that the individual response of 5 subjects is shown. However, more than 5 individual data lines are shown in figure 1.

• L216 – figure 2. It seems that one figure is missing? Figure 2 doesn’t show changes in blood bicarbonate, pH and SID but shows electrolytes?

• Could the authors elaborate on the replicability of the SID measurements. E.g. if you would measure a subject twice, what is the variation in blood electrolytes?

6. PLOS authors have the option to publish the peer review history of their article (what does this mean?). If published, this will include your full peer review and any attached files.

Reviewer #1: **Yes: **Bryan Saunders

Reviewer #2: No

---

## [Author Response · Author response to Decision Letter 0]

19 May 2021

Dear Dr. Sunderland, 

Thank you for allowing us to resubmit our manuscript titled ‘The time to peak blood bicarbonate (HCO3–), pH, and the strong ion difference (SID) following sodium bicarbonate (NaHCO3) ingestion in highly trained adolescent swimmers’ with revisions. We have carefully considered the comments from both reviewers, subsequently making substantial revisions to the manuscript to ensure both reviewers’ comments have been addressed. In summary, we feel the comments have enhanced the strength of the manuscript and we thank the reviewers for both their time and expertise in contributing to the review process. In addition, we can confirm that there are no ethical restrictions on the data set that was used to inform this study, therefore an anonymised Excel spreadsheet has been included with this resubmission.

As requested, we have detailed below a point-by-point response to each of the comments with reference to both the revisions we have made, and the line numbers that the revisions can be found. In the attached revised manuscript, we have highlighted any changes made in red. We hope these responses and subsequent amendments to the manuscript are well received and now meet the required standard. However, we will of course be happy to address any further recommendations that may be required. 

Best regards, 

Author team 

Response to Reviewers’ Comments

Reviewer #1 General Comments

Comment: The authors have done a nice job in determining the difference in time to peak pH vs. HCO3 vs. SID following ingestion of sodium bicarbonate in a group of adolescent swimmers. The methods are generally sound although I have some concerns about the interpretation and the extrapolation of these results. I hope the authors will consider my suggestions.

Response: We thank the reviewer for providing constructive comments to assist our work. Below we have attempted to address the highlighted issues within the manuscript.

Comment: The rationale for using adolescents is rather weak. It feels like the authors are trying to justify using this population for a reason that may not have been the initial reason for using this population. Perhaps it was just a convenience sample? I may be wrong of course, but to me it currently appears this way. Likewise, there is no issue with that in my opinion, but it is best to be clear upfront if this was the case. 

Response: The lead author is currently undertaking his PhD alongside a high-performance pathway swimming club, in which research questions are being raised in response to the competitive practices observed within an adolescent cohort. From a practical perspective, these athletes can be observed partaking in suboptimal supplement practices (including NaHCO3) during competition, which could be detrimental to both health and performance. Based on the latest developments in NaHCO3 research, it is therefore necessary to first identify the pharmacokinetics of HCO3– (and potentially the SID) following ingestion to best inform NaHCO3 supplement practice. Lines 90 – 107 of the introduction has been amended to better justify the identification of time course changes in adolescent blood analytes following NaHCO3 ingestion. We are happy to consider further revisions on this if required.

Comment: The limitation of no performance measure should also be highlighted here as the authors focus somewhat on the suggestion that acidosis may not be limiting to performance, but do not investigate this in any way. 

Response: Upon review of our manuscript, we understand that discussing the possible ergogenic effects may be misleading based upon the contents off the study. We have therefore made attempts to temper our arguments regarding the applicability of a time to peak SID approach in practice.

Comment: I believe the authors must temper statements, extrapolations and conclusions to the sample population used here (adolescents) since this may at least in part explain some of the differences shown here compared to other literature.

Response: The authors agree with this comment and have attempted to make this clearer throughout the manuscript.

Reviewer #1 Specific Comments:

Line 42: In my opinion, the word “adequate” here undersells the strength of evidence regarding SB supplementation and performance. [See line 56 where the authors themselves state “strong ergogenic benefit”]

Response: We have updated these statements on lines 39 and 53 to better reflect the strength of NaHCO3 supplementation.

Line 65: Interestingly, 60 min post-supplementation in the data of Oliveira et al. showed only a 69% likelihood of having increased >5 mmol/L of HCO3, perhaps supporting these data?

Response: Whilst this interesting data certainly supports the lack of blood HCO3–difference found 60 min post-ingestion in the Boegman et al. study, we felt that adding the Oliveira et al. data here would have required further discussion point that would have significantly enhanced the word count of the present introduction. 

Line 76-77: “A concomitant increase in muscular Cl– uptake is also observed [27,29,30]” – Following SB supplementation? Please clarify.

Response: Line 76 has been updated to clarify this change.

Line 97-98: This statement is incorrect. Please see https://doi.org/10.1123/ijsnem.2020-0031 as at least one example. Interestingly, this study also showed low bicarbonate increases at a standardised timepoint in adolescents (albeit 90 min post-supplementation). This study does somewhat try to report some individual variation which might help the authors here.

Response: Thank you for bringing this study to our attention, we were unaware of this publication at the initial time of writing. Lines 90 – 107 and the discussion section (completely reworked) have been amended based upon these findings.

Line 106: “adult-like NaHCO3 absorption” sounds very odd. I suggest rewording.

Response: This sentence has now been removed from the manuscript.

Lines 114-119: While this is interesting information and the sample is clearly top-class, is this necessary here since there are no performance measures? Does being an athlete change the variables measured here?

Response: Though this did not change the variables being measured, we included this information to better inform future studies of our participant sample. Furthermore, we hoped that that this information would add justification for the lack of placebo/control comparison since it was not possible to remove all participants from an additional training session in the short time between testing and British Championships (selection for junior international swimming squads). The reworked discussion section also references the potential early maturation of our participant sample, in which this data may be useful for comparison with adults and other adolescent cohorts.

178-181 (and 144-146): Did the authors also perform this analysis for the group as a whole? Why would the authors believe creatine and beta-alanine supplementation would influence the responses measured here? Certainly, in my experience working with both supplements, they do not affect any of these measures at rest.

Line 245-247: This is very surprising considering my previous comment. Again, I would like to know the author’s rationale for separating these two groups. Was this decision a priori?

Line 288-291: I would be intrigued to know the authors speculation as to why this occurred. Previous work with beta-alanine and sodium bicarbonate supplementation (https://pubmed.ncbi.nlm.nih.gov/21407127/) has not shown beta-alanine to blunt bicarbonate increased with sodium bicarbonate. I cannot think of a mechanism via which beta-alanine supplementation would blunt these changes. Similarly, I am unaware of why creatine supplementation would either. I think it is important to provide some physiological and mechanistic reasoning and speculation as to why this difference occurred.

Response: All swimmers were included in the main group analysis and therefore the statement regarding subgroups has been removed from lines 142 – 144. The authors did not expect creatine or beta-alanine ingestion to alter resting blood samples following NaHCO3 ingestion, but compared groups based upon a recommendation from a previous peer review. Since a statistical difference between groups was observed, we felt compelled to discuss these findings in adolescents. However, it has now been brought to our attention that an error was made in the statistical analysis of these groups and, as expected, no differences were found. The authors would like to praise the peer review process, and we have now made significant changes to the results and discussion sections.

Line 181-182: “Additional effect sizes were calculated where appropriate using Hedges’ g bias correction [41].” – Could the authors please specify exactly where and why it was considered appropriate.

Response: The Hedge’s g correction was used based on the small sample size (n < 20) of our study. We have amended lines 181 – 182 to make this clear to the reader.

Line 183-184: Coefficient of variation of what? Please specify here.

Response: Lines 183 – 184 have been amended to explain that this analysis was carried out for each blood analyte.

Line 189-190: Despite no statistical significance, mean differences of as much as 33 min are fairly large in my opinion. It will be important to discuss this in my opinion.

Response: This was the key finding from the results, however, the authors agree that this may have been overlooked somewhat in the discussion. Large parts of the discussion have been rewritten to reflect the importance of this outcome.

Line 269-271: How many individuals showed a rapid decrease following peak HCO3? And what are the authors considering a rapid decrease here?

Response: Based on the vagueness of this sentence, we have amended lines 291 – 294 to include the number of participants that spent a particular amount of time (>60 min, 30 – 60 min, 0 min) each participant spent above the 5 – 6 mmol∙L-1.

Line 271-273: Certainly, in the adolescent study here this might be the case. Please restrict these claims to the population employed.

Response: The interpretation of the study results has been tempered to an adolescent cohort in lines 294 – 296.

Line 287-288: Is the bicarbonate absorbed?

Response: This discussion point has been moved later in the manuscript, but has been reworded to suggest ‘NaHCO3 uptake’ rather than ‘absorption’ on line 358. 

Line 289: Although technically correct that creatine could increase buffering, its actual contribution to buffering capacity is extremely low. I suggest not calling them both buffering supplements.

Response: The authors agree with this statement; however, this section has been removed based upon the co-ingestion of supplements no longer being a feature of this manuscript.

Line 302-304: I believe there may be, as mentioned previously [Comment Line 97-98].

Response: We have attempted to provide an explanation for the differences between adults vs. adolescent blood analyte responses in lines 318 – 343 and 381 – 392.

Line 315-316: The authors cannot state this since they made no comparison to not ingesting a meal. Please modify this statement, perhaps indicating simply that incidence and intensity of side-effects was low.

Response: Thank you for this recommendation, we have amended lines 374 – 375 to include this more appropriate statement.

Line 317: What is the definition of “minor” here?

Response: We based this definition on the participants recording less than 5/10 on the visual analogue scales. However, upon reflection, we understand that some may interpret a 4/10 rating as a ‘moderate’ severity. This statement has therefore been amended to state that the participants rated the severity at “less than 5 out of 10” on line 376. 

Line 317-319: Was the food intake prior to supplement ingestion recorded? This could provide useful and important information. Particularly when comparing the two groups separated according to creatine and beta-alanine co-supplementation.

Response: Food intake was recorded, and this information has now been presented in the methods section (Table 2, line 148). A correlation analysis has also been conducted upon recommendation from reviewer 2 and this has been included in the results section (lines 251 – 272) and referred to during the discussion.

Line 319: What are the authors referring to here with “This”? The meal in general or the fact it was a high CHO meal?

Response: This was a reference to the co-ingestion of a meal. A comparison could have been made to the similarity in CHO content between the present study and Carr et al. (1.2 vs. 1.5 g/kg BM CHO); however, each participant had a different fat and protein intakes. We decided not to make the comparison between CHO content within the manuscript. Line 377 – 379 have now been amended to make “this” clear to the reader.

Line 338-339: “Stark difference” to what? Between sessions? To data from reference 21?

Response: This statement was meant in reference to the difference in reliability outcomes (excellent vs poor) found within Boegman et al. and Oliveira et al. Lines 404 – 406 have been reworded to make the differences between studies clearer.

Line 338-340: Could the authors please elaborate why they believe venous vs. capillary blood samples might lead to different time course responses. Likewise, why would the different time frames (180 vs 240 min) affect the repeatability of the two studies?

Response: Based upon the discussion points in Oliveira et al., it appears as though the sensitivity/accuracy of venous blood samples were capable of detecting fluctuations across the sampling time frame. Whilst this is not thought to affect the time course response, the long-lasting plateaus mentioned in this study could have meant that a “true peak” could have been identified anywhere between 20 – 240 min. A sampling time frame of shorter duration would therefore have shortened the window available for this fluctuation to occur, and therefore reduced the technical error between trials. We acknowledge that these are strengths of the Oliveira et al. study as opposed to criticisms, however, a venous sampling approach with a 4-hour window may not always be an appropriate strategy for athletes. Although this is an interesting avenue of discussion, we decided not to elaborate on this topic in lines 404 – 411 since this is an area of research that requires much further investigation

Line 338: This should be changed to “one hour following a standardised meal” since they were not ingested together.

Response: This statement in lines 402 - 402 has now been amended to accurately represent the NaHCO3 ingestion conditions of Oliveira et al. study.

Line 351-357: I believe some context needs to be provided. Could the absolute dose provided have contributed somewhat? Mean 20 g (0.3 * 65.3 kg) vs 23 g in reference 22? The authors themselves also speculate that co-ingestion of supplements contributed somewhat to the pharmacokinetics. Additionally, these data here are in adolescents and not in adults which might also have modified the results. Thus, some of these statements should be tempered and contextualised. In fact, there is very similar data here to that shown in reference 22 (of which I am a co-author). However, I believe conclusions here should be restricted to the adolescent athlete.

It is also important to consider that the study of Oliveira performed robust statistical modelling of the data, providing statistical likelihoods and probabilities that bicarbonate increases were above +5 and +6 thresholds. This is entirely different to visually stating some people do not always stay above this threshold for prolonged periods. Again, I urge some tempering of statements because the analyses here were entirely different.

Line 356: Since there are some differences in opinion between groups, I would suggest the authors make it clear that it is they specifically making this recommendation here, and not the entire scientific community (e.g. Due to the findings of the current study, we recommend that the individual time point of peak alkalosis…”). Certainly, this intriguing question cannot be conclusively answered either way right now until more work is performed.

Line 356: Since there are some differences in opinion between groups, I would suggest the authors make it clear that it is they specifically making this recommendation here, and not the entire scientific community (e.g. Due to the findings of the current study, we recommend that the individual time point of peak alkalosis…”). Certainly, this intriguing question cannot be conclusively answered either way right now until more work is performed.

Response: The majority of the discussion section has been restructured and reworded in an attempt to temper the statements made in the original manuscript. We have provided more discussion surrounding the potential differences in adult vs adolescent physiology that may have affected our results, therefore limiting our claims to the population that was studied. We accept reviewer 1’s concerns and hope that the amended manuscript has satisfactorily addressed the above issues. However, we are happy to further amend based upon recommendation.

Line 369-371: “Given that the effects of acidosis on exercise performance are controversial, this finding suggests that using a time to peak SID approach could be a more appropriate NaHCO3 ingestion strategy in practice.” – I don’t believe that these data suggest this at all since you did not measure exercise performance and compare peak SID to peak HCO3. I suggest this statement be removed.

Response: This statement has been tempered to suggest that a time to peak SID approach should be tested versus a time to peak HCO3– approach to determine whether a difference exists in performance (lines 417 – 420).

Line 371-372: This is worded rather oddly (“has an any further ergogenic benefits on exercise performance”). Please consider rephrasing. Likewise, further ergogenic benefit compared to what? Please specify.

Response: We have reworded this sentence in lines 418 – 420 to state: “future studies should seek to identify NaHCO3 dosing strategies based upon time to peak SID could therefore provide a more optimal approach to supplementation compared to strategies based upon an increase in blood HCO3–“.

Line 373-374: I suggest the authors make it clearer and isolate this only to the adolescent group studied here.

Response: This has been attempted throughout the manuscript.

Figure 1. The quality is quite poor and it is difficult to make out the individual timelines (though this may just be due to the uploading process on the journal platform). Perhaps it is just the quality of the figure as it currently stands, but I am struggling to make out many individuals who reached peak HCO3 increases followed by quick decreases (Line 269-271). Could the authors identify individuals consuming other supplements and those not?

Response: Based on this feedback, we have changed the figures to represent the vastly different time course changes in HCO3– and the SID of four different participants. 

Reviewer #2: Newbury et al. rigorously investigated the pharmacokinetics of the increase in strong ions following ingestion of 0.3 g/kg BW sodium bicarbonate in highly trained swimmers. This research group has performed some innovative research on this topic during the last years and is considered as world-leading in the domain of sodium bicarbonate ingestion to improve exercise performance. The authors should be congratulated for using a highly ecological valid design making the study applicable for athletes. My primary concerns are related to the power of the study and to the validity of the measurements obtained.

Response: The authors would like to thank reviewer 2 for their kind words and feedback. We would also like to extend our gratitude for noticing a key error in statical analysis (using a paired samples t-test on independent groups) that has completely reshaped the discussion following the study results. Each of the below comments has been addressed to a level that we deemed satisfactory; however, we are open to further recommendations in order to strengthen the robustness of this investigation.

L130: The authors try to perform the study as realistic as possible. However, all measurements are performed in a rested state. It would be interesting to investigate the impact of a warming-up on the pharmacokinetics of the strong ion increase.

Response: The inclusion of a warm-up has recently been shown to have a significant effect on the time and magnitude of HCO3– following NaHCO3 ingestion (Gurton et al., 2021), therefore it certainly would be interesting to see the possible changes that occur in the SID. The next step following this research is to investigate the time to peak SID approach on exercise performance (which is currently in progress – COVID dependant), where the effects of a warm-up would become apparent. However, a pre-determined time to peak is first required at rest prior to this approach being used in a performance situation. This highlights a limitation of personalised dosing strategies since further testing may be necessary in practice, which in turn increases the logistical and financial burden on the swimmers/swimming club. Though we could not find a suitable place to include this discussion point, we have made a (very small) reference that warm-up may have an effect in line 411.

L135-137: Did the authors perform some correlation analyses to assess whether the macronutrient composition of the pre-exercise meal is related to the pharmacokinetics of the increase in SID?

Response: We have included a correlation analysis based upon your request. This has been included in the results section on lines 251 – 271, and used for discussion points later in the manuscript.

L171: A lot of results were almost significant (p values between 0.05 and 0.10) and effect sizes were quite large. Did the authors perform an a-priori power analysis? I am afraid that a lot of statistical effects failed to reach significance as the power of the study was too low?

Response: Our sample size was determined by the availability of highly trained adolescent swimmers that volunteered and provided parental consent to participate in this research (n = 12). This sample size was deemed appropriate based upon previous NaHCO3 dose response research (Boegman et al. pilot study, n = 8; Gough et al., n = 16; Jones et al., n = 18; Oliveira et al., n = 13), especially when considering the novelty of the participant cohort. Though we did not perform an a-priori power calculation, Jones et al. (2016) stated “based on an a priori power calculation (using Ducker et al., 2013); a minimum of 12 participants were required to achieve 95% power at p < .01”. Nonetheless, the authors agree that the statistical power of the study may have been too low, which prompted the inclusion of bias corrected (Hedges g) effect sizes to discuss changes on most occasions. We also attempted to include as much individual data as possible to be transparent with our results, since we understand that there is a possibility of type I and type II errors occurring with a sample size of n = 12. 

L178: Why did the authors perform a paired t-test and not an unpaired t-test when comparing participants that co-ingested other supplements and those who did not?

Response: This was a genuine error that has not been detected on peer review on multiple occasions. We would like to thank the reviewer for bringing this to our attention since this has significantly altered the outcome of the supplement vs. non-supplement data. Based on these results, we have significantly altered the content included within the results and discussion sections.

L198 – figure 1: In the legend of figure 1 is indicated that the individual response of 5 subjects is shown. However, more than 5 individual data lines are shown in figure 1.

L216 – figure 2: It seems that one figure is missing? Figure 2 doesn’t show changes in blood bicarbonate, pH and SID but shows electrolytes?

Response: The authors made a late change to the figures in order to reduce the number of figures presented in the manuscript. However, this change was mistakenly not reflected in the figure titles. These have now been amended in the results section.

Comment: Could the authors elaborate on the replicability of the SID measurements. E.g. if you would measure a subject twice, what is the variation in blood electrolytes?

Response: Based upon the novelty of this measurement, and the time limitations of the participant cohort, this was unable to be provided for the present study. Previous work by Gough et al. (2017) has shown that the replicability of blood sodium changes following NaHCO3 ingestion is poor, however, which may suggest that blood electrolyte changes may not be reliable. Considering that a time difference may exist between time to peak HCO3– and the SID, this study could act as a catalyst for future research to determine whether an approach based upon the peak SID is actually viable in applied practice. Though we have not discussed this directly, a recommendation for further research regarding repeatability of the data is included in lines 393 – 411.

---

## [Decision Letter · Decision Letter 1]

17 Jun 2021

The time to peak blood bicarbonate (HCO3–), pH, and strong ion difference (SID) following sodium bicarbonate (NaHCO3) ingestion in highly trained adolescent swimmers

PONE-D-21-06382R1

Dear Dr. Sparks,

We’re pleased to inform you that your manuscript has been judged scientifically suitable for publication and will be formally accepted for publication once it meets all outstanding technical requirements.

Kind regards,

Caroline Sunderland

Academic Editor

PLOS ONE

Additional Editor Comments (optional):

Reviewers' comments:

Reviewer's Responses to Questions

**Comments to the Author**

1. If the authors have adequately addressed your comments raised in a previous round of review and you feel that this manuscript is now acceptable for publication, you may indicate that here to bypass the “Comments to the Author” section, enter your conflict of interest statement in the “Confidential to Editor” section, and submit your "Accept" recommendation.

Reviewer #1: All comments have been addressed

Reviewer #2: All comments have been addressed

2. Is the manuscript technically sound, and do the data support the conclusions?

Reviewer #1: Yes

Reviewer #2: Yes

3. Has the statistical analysis been performed appropriately and rigorously? 

Reviewer #1: Yes

Reviewer #2: Yes

4. Have the authors made all data underlying the findings in their manuscript fully available?

Reviewer #1: No

Reviewer #2: Yes

5. Is the manuscript presented in an intelligible fashion and written in standard English?

Reviewer #1: Yes

Reviewer #2: Yes

6. Review Comments to the Author

Reviewer #1: I congratulate the authors on an excellent response to my concerns and suggestions and thank them for treating them with consideration. I believe the updated version of the manuscript reads excellently and has benefitted from the updated analysis and removal of the supplementation subgroups. Congratulations on the work!

I have a couple of minor edits below that the authors may wish to consider, but they are just related to wording or sentence structure.

Line 252: I believe this should read, “The total energy content of the pre-ingestion MEAL also had a moderate…”

Line 327: I believe this should read, “…capacity TO increase blood…”

Lines 411-414: This sentence reads rather awkwardly. Perhaps restructure the sentence to something more along the lines of: “Given that the effects of acidosis on exercise performance are controversial, future studies should seek to identify WHETHER NaHCO3 dosing strategies based upon a peak SID concentration could be more ergogenic than standardised ingestion strategies, or COMPARED TO individualised approaches based upon a time to peak blood HCO3–.”

Reviewer #2: All my previous concerns have been successfully addressed by the authors. I thank the authors for these adjustments.

7. PLOS authors have the option to publish the peer review history of their article (what does this mean?). If published, this will include your full peer review and any attached files.

Reviewer #1: **Yes: **Bryan Saunders

Reviewer #2: **Yes: **Chiel Poffé

---

## [Editor Report · Acceptance letter]

23 Jun 2021

PONE-D-21-06382R1 

The time to peak blood bicarbonate (HCO_3_^–^), pH, and the strong ion difference (SID) following sodium bicarbonate (NaHCO_3_) ingestion in highly trained adolescent swimmers 

Dear Dr. Sparks:

I'm pleased to inform you that your manuscript has been deemed suitable for publication in PLOS ONE. Congratulations! Your manuscript is now with our production department. 

Kind regards, 

on behalf of

Dr. Caroline Sunderland 

Academic Editor

PLOS ONE